# Realization of a crosstalk-avoided quantum network node using dual-type qubits of the same ion species

L. Feng [1,5], Y.-Y. Huang [1,5], Y.-K. Wu [1,2], W.-X. Guo [1,3], J.-Y. Ma [1,3], H.-X. Yang [3], L. Zhang [1], Y. Wang [1], C.-X. Huang [1], C. Zhang [1], L. Yao [3], B.-X. Qi [1], Y.-F. Pu [1,2], Z.-C. Zhou [1,2] & L.-M. Duan [1,2,4] ✉

Generating ion-photon entanglement is a crucial step for scalable trapped-ion quantum networks. To avoid the crosstalk on memory qubits carrying quantum information, it is common to use a different ion species for ion-photon entanglement generation such that the scattered photons are far off-resonant for the memory qubits. However, such a dual-species scheme can be subject to inefficient sympathetic cooling due to the mass mismatch of the ions. Here we demonstrate a trapped-ion quantum network node in the dual-type qubit scheme where two types of qubits are encoded in the $S$ and $F$ hyperfine structure levels of $^{171}Yb^+$ ions. We generate ion photon entanglement for the $S$-qubit in a typical timescale of hundreds of milliseconds, and verify its small crosstalk on a nearby $F$-qubit with coherence time above seconds. Our work demonstrates an enabling function of the dual-type qubit scheme for scalable quantum networks.

As one of the most promising physical systems for quantum computing, trapped ions have demonstrated high-fidelity elementary quantum operations for up to tens of qubits[1–4]. However, it is well-known that the commonly used one-dimensional structure of trapped ions will be restricted to about one hundred qubits in practice[5–7]. To further scale up the qubit number while maintaining the high performance of individual qubits, it is desirable to have a modular design where small-scale quantum computers are assembled together with entanglement efficiently generated between these modules[5,8,9].

One plausible modular scheme for ion trap quantum computers is an ion-photon quantum network, in which distant qubits in individual ion traps are entangled together via photonic links[9–11]. In this scheme, one will repetitively generate ion-photon entanglement on a fraction of ions (called "communication qubits") in each trap. By entanglement swapping of photons from different traps, separated quantum computing modules can thus be connected. Incidentally, this ion-photon quantum network scheme is also compatible with the other schemes like quantum charge-coupled device (QCCD) for their

further scaling[5,8]. For high-fidelity quantum computing, during the generation of ion-photon entanglement in one trap, it is necessary that the randomly scattered photons do not damage the quantum states of the other ions (called "memory qubits") in the same trap. In previous experiments, it is thus common to use two different ion species for the two types of qubits[12–14], so that the transition frequency of the memory qubits is far off-resonant from that of the photons scattered from the communication qubits and hence the crosstalk error is suppressed.

Currently, such a quantum network is mainly limited by the low entanglement generation rate due to the inefficient collection and detection of the photons[3]. Therefore, it is still desirable to increase the qubit number in individual traps as much as possible to minimize the required communication between different traps. However, as the ion number increases, the efficiency of sympathetic cooling will decrease due to the mass mismatch[15]. To solve this problem, recently a dual-type qubit scheme was proposed and demonstrated for crosstalk-avoided sympathetic cooling and single-qubit operations using the same ion

[1]Center for Quantum Information, Institute for Interdisciplinary Information Sciences, Tsinghua University, Beijing 100084, PR China. [2]Hefei National Laboratory, Hefei 230088, PR China. [3]HYQ Co. Ltd., Beijing 100176, PR China. [4]New Cornerstone Science Laboratory, Beijing 100084, PR China. [5]These authors contributed equally: L. Feng and Y.-Y. Huang. ✉e-mail: lmduan@tsinghua.edu.cn

species[16]. Such a scheme using a single ion species for various tasks has now been widely regarded as a promising way for scalable ion trap quantum computing[17]. Despite this progress, to demonstrate the compatibility of the dual-type qubit scheme with the ion-photon quantum network is still experimentally challenging because a long storage time comparable to the entanglement generation timescale will be needed. Actually, even for the commonly used dual-species scheme, this goal was just achieved recently[13].

Here, we generate ion-photon entanglement in a typical timescale of hundreds of milliseconds for the communication qubit, with the memory qubit being protected against the crosstalk error by converting to a different qubit type. We realize long-time storage of the memory qubit by combining spin echoes with the coherent conversion of qubit types, and we compare the storage fidelity with/without ion-photon entanglement generation to explicitly show the negligible crosstalk error. Our work shows the compatibility of dual-type qubit scheme with the ion-photon quantum network, thus makes an important step towards its applications in scalable quantum computing and networking.

## Results

### Scheme

Our experimental setup is shown schematically in Fig. 1. Two $^{171}$Yb$^+$ ions are trapped with a separation of $12\,\mu m$ and can be individually addressed by focused laser beams with a beam waist radius of about $4\,\mu m$. The communication qubit is first pumped to $|^2P_{1/2},F=0,m_F=0\rangle$, and then decays to one of the $^2S_{1/2},F=1$ states through the spontaneous emission of a polarization-entangled photon:

$$|\Psi\rangle = \frac{1}{\sqrt{3}}(|1,-1\rangle|\sigma^+\rangle + |1,0\rangle|\pi\rangle + |1,1\rangle|\sigma^-\rangle) \qquad (1)$$

where we denote the ionic state as $|F,m_F\rangle$ in the $^2S_{1/2}$ levels for simplicity, and $|\sigma^\pm\rangle$ and $|\pi\rangle$ represent the polarization of the photon with respect to a magnetic field $B \approx 5.6\,G$ which sets the quantization axis.

We collect the photon in a direction perpendicular to the magnetic field. From the radiation pattern of the dipole transition, we obtain photon states $|\pi\rangle \to c|V\rangle$ and $|\sigma^\pm\rangle \to c|H\rangle/\sqrt{2}$ where $|H\rangle$ and $|V\rangle$ represent horizontal and vertical polarizations, and $c$ is a small probability amplitude of detecting a photon. After applying a two-tone microwave pulse to turn $(|1,-1\rangle + |1,1\rangle)/\sqrt{2}$ into $|\downarrow\rangle \equiv |0,0\rangle$, we end up with the desired ion-photon entangled state

$$|\Phi\rangle = \frac{1}{\sqrt{2}}(|\downarrow\rangle|H\rangle + |\uparrow\rangle|V\rangle) \qquad (2)$$

where $|\uparrow\rangle \equiv |1,0\rangle$.

The memory qubit can be encoded as an $S$-qubit spanned by $|0\rangle \equiv |0,0\rangle$ and $|1\rangle \equiv |1,0\rangle$. Here we use different notations for the same states to distinguish from those of the communication qubit. In the dual-type qubit scheme, we can also encode as an $F$-qubit spanned by $|0'\rangle \equiv |^2F_{7/2},F=3,m_F=0\rangle$ and $|1'\rangle \equiv |^2F_{7/2},F=4,m_F=0\rangle$. The two qubit types can be coherently converted into each other by focused 411 nm laser and global 3432 nm laser via the intermediate $D_{5/2}$ levels[16] as shown in Fig. 1e. We use bichromatic laser to transfer both qubit levels simultaneously so that the qubit state is insensitive to the laser dephasing. Owing to the large frequency detuning, an $F$-type memory qubit is well protected from the random scattering of 370 nm photons from the communication qubit.

### Ion-photon entanglement

As shown in Fig. 1b and c, we initialize the communication qubit in $|\downarrow\rangle$ by 40 $\mu s$ Doppler cooling and 8 $\mu s$ optical pumping, and then convert it

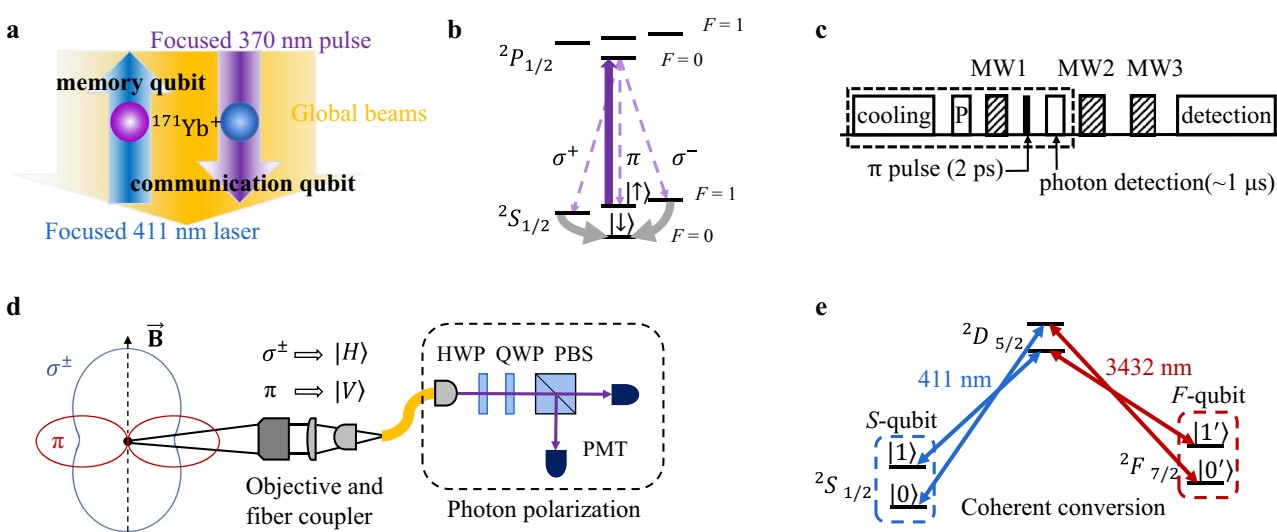

**Fig. 1 | Experimental scheme. a** Two $^{171}$Yb$^+$ ions are separated by $12\,\mu m$ with dual-type qubit encoding. A communication qubit generates ion-photon entanglement in the form of an $S$-qubit under focused 370 nm pulsed laser. A memory qubit stores quantum information in the form of an $F$-qubit, and can be coherently converted to/from an $S$-qubit via a focused 411 nm laser (and a global 3432 nm laser). Both focused laser beams have a beam waist radius of about $4\,\mu m$. Other operations like state initialization and detection are achieved by global laser or microwave beams. **b** Relevant energy levels and **c** experimental sequence for the communication qubit. The dashed box indicates the state initialization and ion-photon entanglement generation cycle including Doppler cooling, optical pumping (P), a microwave $\pi$ pulse (MW1), an ultrafast 370 nm laser $\pi$ pulse and the photon detection by photomultiplier tubes (PMTs). A two-tone microwave (MW2) combines the Zeeman levels of the ion into the $|\downarrow\rangle$ state, and another microwave pulse (MW3) further rotates the qubit between $|\uparrow\rangle$ and $|\downarrow\rangle$ to measure in different basis. **d** Radiation pattern for $\sigma^\pm$ and $\pi$ polarizations with respect to the quantization axis set by the static magnetic field $\vec{B}$. A 0.23-NA objective lens is used to couple the photon into a single-mode fiber, mapping the $\pi$ photon into vertical polarization $|V\rangle$ and the $\sigma^\pm$ photon into horizonal polarization $|H\rangle$. A half-wave plate (HWP) and a quarter-wave plate (QWP) rotate the basis of the polarization qubit, which is then measured by a polarization beam splitter (PBS) and two PMTs. **e** The memory qubit can be protected in the $^2F_{7/2}$ levels as an $F$-qubit ($|0'\rangle$ and $|1'\rangle$), and can be coherently converted to/from an $S$-qubit ($|0\rangle$ and $|1\rangle$) via the intermediate $^2D_{5/2}$ levels. Bichromatic 411 nm and 3432 nm laser are used to transfer the two qubit states simultaneously so that the qubit is not affected by the phase coherence of the laser.

to $|\uparrow\rangle$ by a 12 μs microwave π pulse (MW1). Then we excite the ion to $|^2P_{1/2}, F = 0, m_F = 0\rangle$ by a focused π-polarized 370 nm laser pulse in 2 ps[18] (see Methods). The ion decays to the $^2S_{1/2}, F = 1$ levels in a typical timescale of $1/\Gamma \approx 8$ ns to produce the entangled state in Eq. (1). We repeat this ion-photon entanglement generation cycle (the dashed box in Fig. 1c) until a photon is detected in a time window of 60 ns (although it takes about 1 μs to transfer the data).

Conditioned on the collection of the photon perpendicular to the quantization axis in Fig. 1d, we get an entangled state $\frac{1}{\sqrt{2}}[\frac{1}{\sqrt{2}}(|1, -1\rangle + |1,1\rangle)|H\rangle + |\uparrow\rangle|V\rangle]$. However, note that the time when the spontaneous emission occurs is uncertain in the 60 ns time window, which means that the relative phases between these Zeeman levels cannot be determined in advance, and distribute in a range of $\Delta\omega_{Zeeman}\Delta T \approx 3$. In the experimental sequence, we compensate this phase by feedforwarding the detection time of the photon into the phase of the two-tone microwave pulse (MW2 in Fig. 1c, 13 μs) so that an entangled state in Eq. (2) is obtained independent of the time of the spontaneous emission.

We then bound the entanglement fidelity by rotating the spin and the photon in different bases[19] as shown in Fig. 2. In the diagonal basis, we measure the correlation between the spin and photon as $P(\uparrow|V) = 0.964 \pm 0.007$, $P(\downarrow|V) = 0.036 \pm 0.007$, $P(\uparrow|H) = 0.077 \pm 0.010$ and $P(\downarrow|H) = 0.923 \pm 0.010$ from about 1000 trials with the error bars computed from a binomial distribution. Then we consider the off-diagonal basis where the spin and the photon are both rotated by π/2 pulses with a relative phase of φ. For the spin, the rotation is achieved by the MW3 pulse in Fig. 1c, and for the photon we rotate the

polarization direction by the half-wave plate in Fig. 1d. As shown in Fig. 2c, when we scan the phase of the MW3 pulse, we observe a sinusoidal oscillation in the conditional probabilities $P(\tilde{\uparrow}|\tilde{V})$ and $P(\tilde{\uparrow}|\tilde{H})$ (and thus $P(\tilde{\downarrow}|\tilde{V})$ and $P(\tilde{\downarrow}|\tilde{H})$ as well). At the optimal φ which gives largest $P(\tilde{\uparrow}|\tilde{V}) - P(\tilde{\uparrow}|\tilde{H})$, we obtain $P(\tilde{\uparrow}|\tilde{V}) = 0.959 \pm 0.008$, $P(\tilde{\downarrow}|\tilde{V}) = 0.041 \pm 0.008$, $P(\tilde{\uparrow}|\tilde{H}) = 0.091 \pm 0.013$, $P(\tilde{\downarrow}|\tilde{H}) = 0.909 \pm 0.013$, as shown in Fig. 2b. From these results, we bound the ion-photon entanglement fidelity[19] as $F \geq \frac{1}{2}[P(\uparrow, V) + P(\downarrow, H) - 2\sqrt{P(\downarrow, V)P(\uparrow, H)} + P(\tilde{\uparrow}, \tilde{V}) + P(\tilde{\downarrow}, \tilde{H}) - P(\tilde{\uparrow}, \tilde{H}) - P(\tilde{\downarrow}, \tilde{V})] = (88.0 \pm 1.0)\%$, which is far above the classical threshold of 0.5 and thus verifies the entanglement. We measure an entanglement generation rate $r \approx 5\,s^{-1}$, which is consistent with our repetition rate of $1.6 \times 10^4\,s^{-1}$, the solid angle $\Delta\Omega/4\pi \approx 1.3\%$ of the 0.23-NA objective, the probability 2/3 to project to the maximally entangled state of Eq. 2, the 15% fiber coupling efficiency, the 30% detection efficiency of the photomultiplier tube (PMT) and the 85% transmission efficiency for all the other optical elements.

As for the error budget, we estimate 2% infidelity from optical pumping and state detection of the ion, 3% from imperfect π polarization of the 370 nm laser pulse which leads to excitation to $|^2P_{1/2}, F = 1, m_F = \pm 1\rangle$ states, 2% from the misalignment of the objective to collect the photon, and 2% from the dark count of the PMT. Another 6% infidelity can come from the imperfect conversion from $(|1, -1\rangle + |1,1\rangle)/\sqrt{2}$ to $|0,0\rangle$, whose contributions include 4% from the 2 ns jitter time of the control system, as well as 2% from the dephasing between the two Zeeman levels during the microwave pulse. More details about the ways we calibrate the experimental system and estimate the error can be found in Supplementary Note 1.

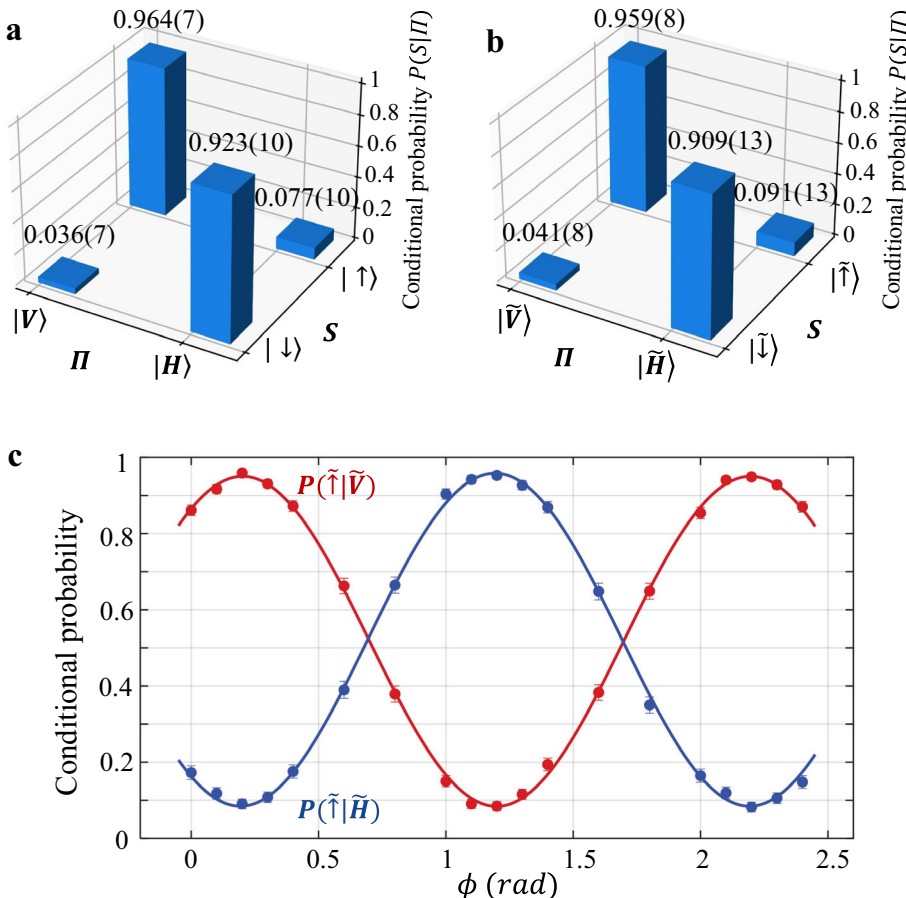

**Fig. 2 | Fidelity bound of ion-photon entanglement. a** Conditional probabilities $P(S|\Pi)$ in the diagonal basis with $S = |\uparrow\rangle, |\downarrow\rangle$ and $\Pi = |H\rangle, |V\rangle$. **b** Conditional probabilities in the off-diagonal basis under optimal phase φ. Combining the two basis, we bound the entanglement fidelity by $F \geq 88.0\%$. **c** We apply π/2 pulses to both the ionic and the photonic qubits with their relative phase φ controlled by that of MW3 in Fig. 1. We scan φ to obtain the highest conditional probabilities $P(\tilde{\uparrow}|\tilde{V})$ and $P(\tilde{\downarrow}|\tilde{H})$ as shown in (**b**). All error bars represent 1 s.d.

## Long-time quantum memory of dual-type qubit

Given the above entanglement generation rate, we expect a typical timescale of hundreds of milliseconds to obtain an ion-photon entanglement. Therefore, we want to extend the storage lifetime of the dual-type qubit to a similar timescale to demonstrate its compatibility with the ion-photon quantum network. Since our dual-type qubit is encoded in the hyperfine clock states, it is largely insensitive to the fluctuation in the magnetic field or the AC Stark shift of the scattered laser beams. However, since finally we will bring the $F$-qubit back to the $S$-type for detection, even a small frequency mismatch in the bichromatic laser on Hertz-level from the qubit transition frequency can accumulate into considerable phase shift during the sub-second storage. Therefore, we apply spin echoes by converting the qubit to the $S$-type, performing a microwave $\pi$ pulse, and then converting it to the $F$-type again, as shown in the experimental sequence in Fig. 3a. In this way, the phase shift on the two qubit states cancel with each other. We further choose to use two spin echoes for a balance between the imperfect qubit-type conversion and the cancellation of the phase errors.

To benchmark the storage fidelity of the memory qubit, we initialize it as an $S$-qubit in one of the six mutually unbiased bases (MUBs)[20] $|0\rangle$, $|1\rangle$, $|+\rangle$, $|-\rangle$, $|L\rangle$, $|R\rangle$ with equal probability, convert it to the $F$-type, store for a total time $T = 4\tau$ separated by two spin echoes, and finally move it back to the $S$-type to measure the storage fidelity. For a fair comparison with the later experiment, here we use an EMCCD to measure the memory qubit state instead of the PMT. As shown in Fig. 3b, the fidelity oscillates with the storage time, and can be well fitted by a single-frequency noise at around $(57.3 \pm 0.1)$ Hz and a dephasing time $T_2 = (2.8 \pm 0.7)$ s[21,22] (see Methods). This frequency varies slightly when we repeat the experiment and its source is still under investigation. For a typical storage time $T = 200$ ms, the storage fidelity for the 6 basis states are summarized in Fig. 3c, which gives an average fidelity of $(83.9 \pm 0.4)\%$ where the error bar represents one standard error for the average of the total 9000 trials. The infidelity contains three round-trips of the qubit-type conversion with 3% error each (see Methods), 2.5% from state preparation and detection using an EMCCD, and $(1 - e^{-T/T_2})/2 \approx 3\%$ from the decoherence during long-time storage.

## Crosstalk-avoided quantum network node

With the ion-photon entanglement and the long-time storage of the dual-type qubit demonstrated separately, now we can combine them together to establish a crosstalk-avoided quantum network node. The experimental sequence is basically to insert the ion-photon entanglement generation on the communication qubit (Fig. 1c) into the storage time $4\tau = 200$ ms of the memory qubit (Fig. 3a), as shown in Fig. 4a. We repeat the entanglement generation attempts until a photon is collected or the storage time window is used up. For the simplicity of the control sequence, here we only attempt to generate entanglement in the middle $2\tau = 100$ ms, which corresponds to a success probability of about 40% for each trial given the generation rate of 5 s⁻¹. Note that this nonunity success rate does not affect our estimation of the crosstalk error, which is proportional to the number of generated photons, not that of the collected ones.

At the end of the experimental sequence, we use an EMCCD to measure the states of the two qubits simultaneously. Again, we prepare the memory qubit in one of the six MUBs randomly. For 1800 successful trials when the communication qubit and the photon are measured in the diagonal basis (with the conditional probability shown in Fig. 4b), we get about 300 trials for each MUB and an average storage fidelity of $(83.6 \pm 0.8)\%$. This is consistent with the storage fidelity $(83.9 \pm 0.4)\%$ without ion-photon entanglement generation within the error bars, thus proves a negligible crosstalk error. In comparison, without the dual-type qubit encoding, the memory qubit would have experienced much larger crosstalk error which would have

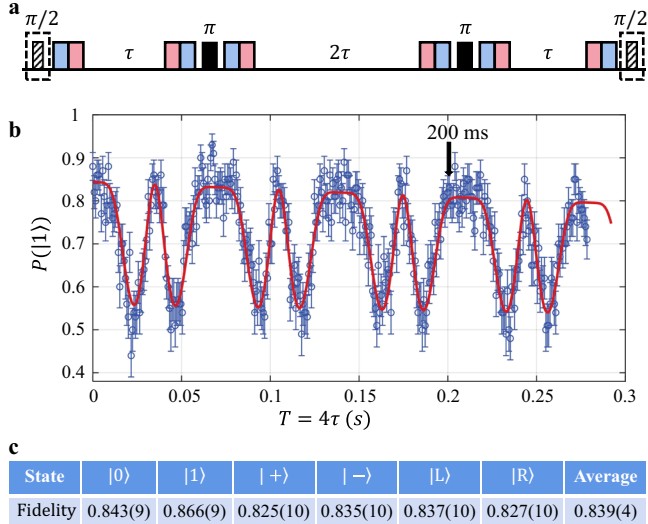

**Fig. 3 | Storage fidelity of memory qubit. a** Experimental sequence. The memory qubit is initialized in one of the mutually unbiased bases (MUB) as an $S$-qubit by suitable microwave pulses (indicated by the dashed box). Then we convert it to $F$-qubit by 411 nm (blue bars) and 3432 nm (red bars) laser. The total storage time $T = 4\tau$ is divided into three segments separated by two spin echoes. To keep the phase coherence with the initial $\pi/2$ pulse and to cancel the small frequency mismatch between the bichromatic laser and the qubit, we always bring the qubit back to $S$-levels for the spin echo, such that $|0\rangle$ and $|1\rangle$ accumulate equal phases during the storage. After a final microwave pulse, ideally the memory qubit should be in the bright state $|1\rangle$. **b** Storage fidelity for an initial $|+\rangle$ state versus storage time $T$. The observed curve can be well fitted by a single-frequency noise at $(57.3 \pm 0.1)$ Hz and a dephasing time $T_2 = (2.8 \pm 0.7)$ s. **c** We choose $T = 200$ ms and measure the storage fidelity for the MUBs to obtain an average fidelity of $(83.9 \pm 0.4)\%$ for the total 9000 trials. All error bars represent 1 s.d.

been detected under our experimental precision, and would have even exceeded our overall measured storage infidelity of 16% with the technical errors included (see Supplementary Note 2 for details). We also repeat the measurement for the communication qubit and the photon in the off-diagonal basis in Fig. 4c. This gives us a bound for entanglement fidelity $F \geq (84.8 \pm 0.9)\%$. The additional 3.2% infidelity compared with the entanglement fidelity in Fig. 2 mainly comes from the state detection error when switching from the PMT to the EMCCD (see Supplementary Note 1 for details).

## Discussion

To sum up, we have demonstrated the long storage lifetime of the dual-type qubit scheme and further show its compatibility with an ion-photon quantum network with negligible crosstalk error, which is an enabling function of the dual-type qubit scheme in large-scale quantum computing. In previous experiments, if the same ion species is used for the communication and the memory qubits (e.g. Ref. 23), one has to first generate ion-photon entanglement and then reset the memory qubit for the following quantum operations, so as to circumvent the problems of storage lifetime and crosstalk error. Here with the dual-type qubit scheme, we are able to protect the stored quantum information while generating ion-photon entanglement, which makes a crucial step toward the future modular quantum computers and networks. On the other hand, compared with the recent demonstration using the dual-species scheme[13], our scheme requires only a single ion species with simpler control techniques, thus can be advantageous when scaling up to larger qubit numbers. Admittedly, this simplification in hardware is achieved at the cost of more complicated software to convert the qubit type back and forth for the storage and readout, whose fidelity still needs to be improved.

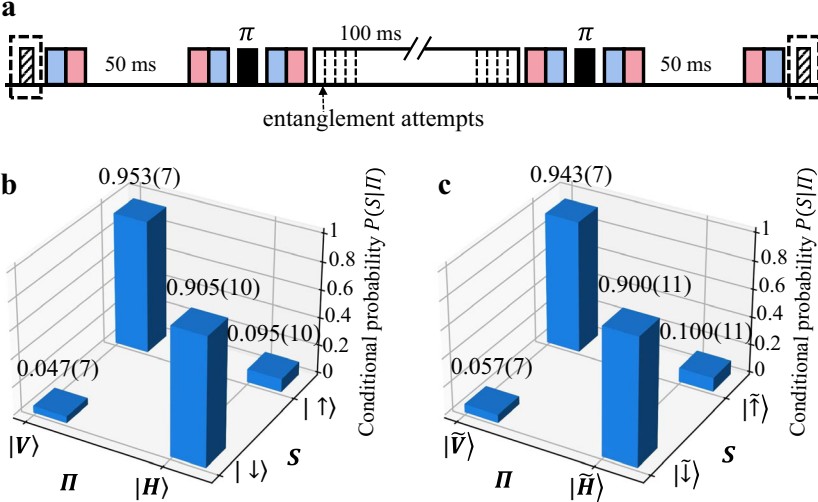

**Fig. 4 | Demonstration of a crosstalk-avoided quantum network node. a** We follow a similar experimental sequence as Fig. 3a with $T = 200$ ms, but keep attempting to generate ion-photon entanglement in the middle 100 ms on the communication qubit. After the sequence, the states of the two ions are measured by an EMCCD camera. The conditional probabilities for ion-photon entanglement in the diagonal and off-diagonal bases are displayed in (**b**) and (**c**), respectively, with an entanglement fidelity $F \geq 84.8\%$, and we obtain an average storage fidelity of $(83.6 \pm 0.8)\%$ for the memory qubit from 1800 trials. All error bars represent 1 s.d.

Also, our scheme needs individual addressing with focused laser beams to convert the desired qubits selectively. This is already available in many current multi-ion quantum computers for individually addressed quantum gates[24–26].

The current storage infidelity of the memory qubit is about 16% over the 200 ms storage time, which must be significantly improved for the future fault-tolerant quantum computing. As mentioned above, among the 16% infidelity, only about 3% comes from the measured coherence time, while the dominant sources are the conversion infidelity between the $S$-qubit and the $F$-qubit, and the SPAM error using EMCCD. Compared with our previous work, here our round-trip qubit-type conversion fidelity is lowered by about 2%[16]. This is because we only perform sympathetic Doppler cooling during the long time storage while in Ref. 16 we further perform sideband cooling. An analysis about the effect of the thermal phonon number on the conversion fidelity can be found in Supplementary Note 3. This can be improved in the future by pausing the entanglement generation sequence once every tens of attempts to provide sideband cooling, or we may add more ions into the system to provide sympathetic sideband cooling continuously. As for the SPAM error from the EMCCD detection, it can be improved by electron shelving[16,27–30]. To incorporate electron shelving into the dual-type qubit scheme, we may choose to shelve to other Zeeman levels of $F_{7/2}$ not used for encoding qubits, or we may use the sequence of "global 3432 nm $\pi$ pulse"-"individual 411 nm $\pi$ pulse"-"global 3432 nm $\pi$ pulse" to selectively convert qubit types for the desired ions, given that the conversion fidelity can be improved as mentioned above.

Note that in this work we only verify a vanishing crosstalk error at the precision of about 1% by computing the change in the storage fidelity with/without ion-photon entanglement generation, although theoretically the crosstalk is far below the fault-tolerant threshold (see Supplementary Note 2). To actually confirm a crosstalk below the threshold of, say, $10^{-4}$, a much higher measurement precision of the storage fidelity would be needed. From a scaling of $1/\sqrt{M}$ for the statistical fluctuation, we estimate a total trial number of $M = 10^8$ or a total experimental time on the year level, which will be highly demanding for the stability of the system. A more practical scheme would be to have a large ion crystal with smaller ion distances to

generate ion-photon entanglement simultaneously, and to measure the total crosstalk error on a memory qubit at the center of the crystal. Also note that this crosstalk error is more like a memory error rather than the gate error or the measurement error in quantum error correction, and is thus less severe for fault-tolerance[31]. Nevertheless, it is always desirable to achieve and to verify lower crosstalk errors to save the overhead for large-scale quantum error correction.

As for the entanglement fidelity of the communication qubit, our current scheme to generate entanglement is based on our laser and magnetic field orientations. In the future we may upgrade the laser configuration to allow simpler entanglement generation scheme without the need to combine the amplitude on two Zeeman levels coherently[23], then the jitter time of the control system and the dephasing of the Zeeman levels will have weaker effect (at least in the diagonal basis). Using electron shelving can also help to reduce the state detection error as stated above. To further enhance the entanglement generation rate, we can use a higher NA objective, increase the fiber coupling rate, and use high-efficiency detectors. There is also room to shorten the sequence length for each entanglement generation attempt with careful design.

## Methods
### Ultrafast 370 nm $\pi$ pulse
We use a 2 ps ultrafast 370 nm laser pulse to pump the communication qubit from $|^2S_{1/2}, F = 1, m_F = 0\rangle$ to $|^2P_{1/2}, F = 0, m_F = 0\rangle$. The pulsed laser comes from a mode-locked Ti:sapphire laser with 739 nm central wavelength, and is frequency-doubled to 369.5 nm by a second harmonic generator system (SHGS). Before the SHGS, an electro-optic modulator is used to pick a single pulse from the 76 MHz pulse train, giving an extinction ratio of about $10^5$:1. More details can be found in our previous work[18].

To calibrate the pulse area, we apply a microwave $\pi$ pulse to the state after spontaneous emission, so as to convert the population from $|^2S_{1/2}, F = 1, m_F = 0\rangle$ to $|^2S_{1/2}, F = 0, m_F = 0\rangle$, which appears dark whereas the other states in the $^2S_{1/2}, F = 1$ manifold appear bright in our detection scheme. Ideally, if the laser has a pulse area of $\pi$, the probability of getting a bright state in the end will be 2/3. Otherwise, we expect a bright state probability $(2/3)\sin^2(\theta/2)$ where $\theta$ is the pulse area. As

shown in Supplementary Figure 1 in Supplementary Note 4, we fit this curve and obtain a laser $\pi$ pulse with the excitation probability to the $|^2P_{1/2}, F = 0, m_F = 0\rangle$ state reaching $P_e \approx 99\%$.

**Fitting single-frequency noise on memory qubit**

Consider a single-frequency noise with amplitude $A$, frequency $\omega$ and a random phase $\varphi$ on the qubit frequency[21,22], the Hamiltonian can be written as: $H = A\cos(\omega t + \varphi)\sigma_z/2$. We start from the superposition state $(|0\rangle + |1\rangle)/\sqrt{2}$. After the storage time of $T = 4\tau$ separated by two $\pi$ pulses, we obtain a phase shift between $|0\rangle$ and $|1\rangle$ as

$$\Delta\phi(\varphi) = \frac{A}{\omega}[-\sin\varphi + 2\sin(\omega\tau + \varphi) - 2\sin(3\omega\tau + \varphi) + \sin(4\omega\tau + \varphi)].$$

(3)

Thus the dephasing of the off-diagonal terms of the density matrix is given by

$$\langle e^{i\Delta\phi}\rangle \equiv \frac{1}{2\pi}\int_0^{2\pi} d\varphi e^{i\Delta\phi(\varphi)} = J_0\left(\frac{8A}{\omega}\sin^2\frac{\omega\tau}{2}\sin\omega\tau\right),$$

(4)

and we obtain a fidelity for the stored $|+\rangle$ state as $(1 + \langle e^{i\Delta\phi}\rangle)/2$.

In the experiment, we fit the measured fidelity in Fig. 3b vs. the total storage time $T = 4\tau$ by $F(T) = \{1 + J_0[(8A/\omega)\sin^2(\omega T/8)\sin(\omega T/4)]e^{-T/T_2}\}/2 - c$ where an exponential decay term is used to represent the pure dephasing time $T_2$ and a constant $c$ is subtracted to account for the state preparation and measurement (SPAM) error.

**Coherent qubit type conversion**

We calibrate the coherent conversion fidelity of the dual-type qubit by repeating the $S$-$F$-$S$ round-trip cycles for $N$ times, and fit the decay of the average state fidelity over an MUBs by $F = F_0(1-\epsilon)^N$. As shown in Supplementary Figure 2 in Supplementary Note 5, we extract a SPAM fidelity of $F_0 = 97.0\%$ and the round-trip conversion error of $\epsilon = 3.1\%$.

## Data availability

The data that support the findings of this study are available from the authors upon request. Source data are provided with this paper.

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

## Acknowledgements

This work was supported by Innovation Program for Quantum Science and Technology (2021ZD0301601), Tsinghua University Initiative Scientific Research Program, and the Ministry of Education of China. L.M.D. acknowledges in addition support from the New Cornerstone Science Foundation through the New Cornerstone Investigator Program. Y.K.W. acknowledges in addition support from Tsinghua University Dushi program and the start-up fund.

## Author contributions

L.F., Y.Y.H., Y.K.W., W.X.G., J.Y.M., H.X.Y., L.Z., Y. W., C.X.H., C.Z., L.Y., B.X.Q., Y.F.P, and Z.C.Z. carried out the experiment and analyzed the data. L.M.D. proposed the experiment and supervised the project. Y.K.W., L.F, Y.Y.H., and L.M.D. wrote the manuscript.

## Competing interests

W.X.G., J.Y.M., H.X.Y and L.Y. are affiliated with HYQ Co. L.F., Y.Y.H., Y.K.W., Y.W., B.X.Q., Y.F.P., Z.C.Z. and L.M.D. hold shares of HYQ Co. The other authors declare no competing interests.
