## [Peer Review File · Nature Communications]

Realization of a crosstalk-avoided quantum network node using dual-type qubits of the same ion speciesREVIEWER COMMENTS

Reviewer #1 (Remarks to the Author):

The authors present evidence of generating ion-photon entanglement while preserving the quantum information of other ions in the trap. This is done by using two encodings of the quantum information in the same atomic ion. It relies on the coherent conversion of the atomic ion state from one manifold to another manifold. The research group has previously published on the qubit conversion technique (Nat. Phys. 2022).

Previous work on ion-photon entanglement while preserving the quantum state of the other ions used two-species of ions. The qubit conversion effectively creates two-types of ions, networking ions and memory ions, by choosing different manifolds. This simplifies the experiment since all ions are loaded the same, have the same mass to charge, and share the same laser sources. The normal modes will not depend on the placement of the two-types of ions.

The work is well described, can be reproduced, and the data analysis is correct.

The weakness of the results are the measured fidelities. The authors explain in the conclusion multiple approaches for improvements. As this is the first demonstration of the method, the current fidelities are sufficient for publication. For this method to have significant impact on future quantum computers and networks, the fidelities of storage and a qubit-conversion need to be improved.

Reviewer #2 (Remarks to the Author):

In the manuscript by L. Feng, et al, the authors employ two Yb⁺ ions to demonstrate generation of ion-photon entanglement from one ion while preserving quantum information stored in an adjacent ion serving as a quantum memory. The authors claim low crosstalk achieved by means of encoding of the second “memory” ion in a second qubit encoding utilizing the F-shell states of the Yb ion, as demonstrated by the same group in previous work (reference 16 of the current manuscript). The work addresses an important

problem in trapped-ion quantum information technology, namely the operation of quantum networking protocols in concert with requirements for quantum computation.

While both the ion-photon entanglement fidelity, repetition rate, and preparation/readout fidelity of the memory qubit appear to be substantially lower than in comparable work by Drmota et al also demonstrating ion-photon entanglement in the presence of a long-lived memory using two distinct ion species (ref. 13 of the current manuscript), the novelty of the present manuscript is that the memory ion is encoded in the same physical species which in principle brings benefits for long-term architectures. As such the present manuscript does contribute a meaningful and novel advance, despite the technically lower performance at present. The experiments are nontrivial and challenging.

In general the broad claims of the paper appear sound and well justified. However, my main comments are that multiple claims about the quantitative performance of the demonstrated protocol are insufficiently detailed, and in my view require substantial clarification before the manuscript can be published. This is because these quantitative aspects are key to understanding and evaluating the significant claims of the manuscript. A few major suggestions and questions for clarification that are in my view key to assessing the broad claims of the paper are listed below, together with more minor suggestions for improvements to the text.

(1) The main claim of the paper is that the dual-type encoding scheme permits long-lived memory even during the ion-photon entanglement generation protocol. However, the two Yb ions are spaced by 12 μm , a significant distance, and it is not at all clear from the discussion how severe the crosstalk would be if the same $S_{1/2}$ encoding were used for the memory qubit. Can the authors include a measurement, or at least an estimate, of how severe the crosstalk-induced decoherence would be for the memory ion if a naive encoding in the same state were used, especially in light of the fact that the ions can in principle be more widely separated, e.g. by 100 μm in typical architectures? I.e., why was 12 μm chosen as the ion spacing? Given the $\sim 85\%$ fidelities for the memory reported, it is not clear that a memory ion in the $S_{1/2}$ state would undergo decoherence on this level even in the presence of the entanglement generation. Furthermore, since other trapped-ion systems do

allow multiple photon scattering events on single ions even with other memory ions encoded (e.g. in Quantinuum's systems, simply using some distance between measured and memory ions, e.g. PRA 104, 062440, 2021), this is a crucial comparison to make to understand how much the dual-type encoding really helps in this work. This is doubly important because the encoding used in this work entails significant infidelity in memory encoding/readout (~84% mapping/storage fidelity). As such this seems a critical comparison to assess the value added by the demonstrated mapping.

(2) Almost no detail is given to support the error budget described in the paragraph starting at line 184 — the authors should detail where the estimates for state detection/pumping, polarization, and misalignment come from, in the methods or supplementary information. As stated there is no basis for a reader to assess whether these estimates are meaningful/accurate.

(3) Similarly, in line 275, the 2.6% lower ion-photon entanglement achieved in the presence of the memory ion is explained as due to having switched from PMT to EMCCD — however, it is not clear why this should cause lower infidelity. Can the authors justify this due to higher effective dark count rates in the EMCCD, e.g., or is this due to crosstalk between ions as imaged? To claim that the error is due to the mechanism, the reason for the error should be detailed and at least estimated, or else the authors should revise to state that this is only suspected to be the cause for the increased error.

(4) In line 294, it is stated that the use of single ion entails simpler control techniques. I think this is appropriate with respect to sympathetic cooling and transport — however, with respect to the actual memory encoding and readout, the control is significantly more complex as evidenced in this work. This tradeoff should be acknowledged.

(5) In general, measurement uncertainties, while likely well justified and correctly measured, are not clearly reported. Figs. 2-4 are missing statistical uncertainties on plotted points. No error bar is reported for the 87.9% ion-photon fidelity in line 174. In line 235, it is not entirely clear what is meant by “one standard deviation for the total 9000 trials” — do the authors mean instead the standard error of the average fidelity inferred from these

9000 trials? In general, a more careful treatment of the measurement uncertainties would strengthen the presentation significantly.

A few more detailed minor suggestions for text:

(1) line 21 — replace “ion trap has” with “trapped ions have”

(2) line 42 — replace “scalization” with “scaling”

(3) line 152 — since the fidelity was bounded but not measured by tomography, replace “measure” with “bound”. Similarly in the title to Fig. 2.

(4) Line 266 — replace “300 data” with “300 trials”

Reviewer #3 (Remarks to the Author):

The work reported is a proof of principle demonstration of the use of the dual-type-state-manifold paradigm for generating ion-photon entanglement; this is done while simultaneously storing quantum information in a separate ion of the same species. Performing this operation with low crosstalk on the memory ion, also located in the same ion crystal, is one of the purported advantages of this paradigm, especially in light of the straightforward vibrational mode structure of a same-species ion crystal. Though there are many precursor experiments utilizing metastable states in trapped ions for protected memory, and in using spontaneous emission from ions to create ion-photon entanglement, the particular demonstration is novel in that it combines these two operations in the dual-type paradigm. Its publication should be useful for those in the field; improvements in the experiments must be undertaken before this can be a useful technique in trapped-ion quantum information processing and before a true limit to the dual-type technique can be reported, however, due to the low fidelities obtained here. The major drawback of the current work is that these relatively high infidelities limit the level at which crosstalk can be evaluated. Nonetheless, if this is clearly stated in the manuscript, I believe the work could be published.

Summarizing improvements based on this general point and other detailed suggestions:

1) The authors should, in the discussion, provide an estimate for expected crosstalk during the full experiment (memory during entanglement generation), and then highlight the improvements needed in the basic operations in order to be able to measure either a) crosstalk at this expected level, or if that level is much smaller than typical infidelities suggested for fault-tolerant operation (e.g. 10^{-4}), b) improvements required to get the crosstalk bound below such fault-tolerant infidelities.

2) Wording suggestion for the title: "Realization of a crosstalk-avoided quantum network node using dual-type qubits of the same ion species."

3) Lines 59-61: the motivation for dual-type is a little confusing here when discussing the challenges of dual-species operation--perhaps the authors mean to say that the proportion of ions of a particular species and the ion order are hard to control? On the contrary, it seems that the recent work of Quantinuum has shown that several multi-ion, dual-species crystals of a particular ion order can be accomplished without much difficulty. I would remove this statement and focus on the challenges of cooling all the vibrational modes in a dual-species crystal, as is mentioned in the next sentence.

4) The authors should describe the limitations to crosstalk avoidance presented by the use of focused laser beams for individual addressing, one of the drawbacks of dual-type protocols for qubits in the same ion crystal. The beam waists for the addressing beams quotes here must be comparable to the inter-ion spacing, leading to accidental crosstalk in the mapping operations. This deserves some discussion, as it could present a practical limitation, particularly in terms of attainable trap-potential strength, or to speed if the potential must be weakened to perform this operation.

5) How is state initialization of one ion in a two-ion crystal done here? Are there challenges due to scattered photons affecting the other ion in this case?

6) Why is the qubit brought from the F state to the S state for MW operations? Can these not be done in the F state to avoid the infidelity of two state-mapping operations?

7) Fig 1 suggestion: use the same colors for the laser beams corresponding to each transition in parts a, b, and e to allow easier comprehension by the reader. Also, label the laser beams with wavelengths in part e.

8) It is not stated why the EMCCD-based detection adds more errors--this should be explained, and any fundamental or technical limitations in the number of measurable ions due to this effect should be described.

9) One of the fidelity limitations was due to ion heating since sympathetic cooling was not used. It's not clear that this would be a major limitation unless the ions were heated to a particularly high temperature; what was the heating rate in this experiment, and was it the main limitation to mapping operation infidelity? Or was there a limit fur to heating from the photon entanglement attempts causing recoil? If so, how could this be fixed? (Is a third ion required?) This is not well explained in the paper.

10) The authors discuss a future approach not requiring the combination of two Zeeman levels coherently--was this a limit to fidelity in the current scheme? If so, the authors should describe that. Also, in this same part of the paper, the authors also state that state detection could be improved using electron shelving--how could this be implemented in the current scheme, since the F state is being used and the D state is not long lived? The authors should expand this point with another sentence or two describing how this could be done without involving undue complications.

11) The English should be proofread and edited throughout. Also, the hyphenation breaks across lines are very atypical, e.g. line 287, etc.

If the authors suitably address my concern listed here, and if the other reviewer(s) agree, I believe the work could be published.

Reply to Reviewers

Reviewer #1:

Comment:

The authors present evidence of generating ion-photon entanglement while preserving the quantum information of other ions in the trap. This is done by using two encodings of the quantum information in the same atomic ion. It relies on the coherent conversion of the atomic ion state from one manifold to another manifold. The research group has previously published on the qubit conversion technique (Nat. Phys. 2022).

Previous work on ion-photon entanglement while preserving the quantum state of the other ions used two-species of ions. The qubit conversion effectively creates two-types of ions, networking ions and memory ions, by choosing different manifolds. This simplifies the experiment since all ions are loaded the same, have the same mass to charge, and share the same laser sources. The normal modes will not depend on the placement of the two-types of ions.

The work is well described, can be reproduced, and the data analysis is correct.

The weakness of the results are the measured fidelities. The authors explain in the conclusion multiple approaches for improvements. As this is the first demonstration of the method, the current fidelities are sufficient for publication. For this method to have significant impact on future quantum computers and networks, the fidelities of storage and a qubit-conversion need to be improved.

Reply:

We thank the reviewer for carefully reading through our manuscript and for the positive evaluation that the work is well described, can be reproduced, and that the data analysis is correct. We also agree with the reviewer that the current fidelity still needs to be increased for practical applications, and we have discussed possible directions for future improvement. As for a demonstration of the method, we thank the reviewer for supporting that the current fidelities are sufficient for publication.

Reviewer #2:**Comment:**

In the manuscript by L. Feng, et al, the authors employ two Yb⁺ ions to demonstrate generation of ion-photon entanglement from one ion while preserving quantum information stored in an adjacent ion serving as a quantum memory. The authors claim low crosstalk achieved by means of encoding of the second “memory” ion in a second qubit encoding utilizing the F-shell states of the Yb ion, as demonstrated by the same group in previous work (reference 16 of the current manuscript). The work addresses an important problem in trapped-ion quantum information technology, namely the operation of quantum networking protocols in concert with requirements for quantum computation.

While both the ion-photon entanglement fidelity, repetition rate, and preparation/readout fidelity of the memory qubit appear to be substantially lower than in comparable work by Drmota et al also demonstrating ion-photon entanglement in the presence of a long-lived memory using two distinct ion species (ref. 13 of the current manuscript), the novelty of the present manuscript is that the memory ion is encoded in the same physical species which in principle brings benefits for long-term architectures. As such the present manuscript does contribute a meaningful and novel advance, despite the technically lower performance at present. The experiments are nontrivial and challenging.

In general the broad claims of the paper appear sound and well justified. However, my main comments are that multiple claims about the quantitative performance of the demonstrated protocol are insufficiently detailed, and in my view require substantial clarification before the manuscript can be published. This is because these quantitative aspects are key to understanding and evaluating the significant claims of the manuscript. A few major suggestions and questions for clarification that are in my view key to assessing the broad claims of the paper are listed below, together with more minor suggestions for improvements to the text.

Reply:

We thank the reviewer for carefully reading through our manuscript and for the evaluation that “the work addresses an important problem in trapped-ion quantum

information technology”, “the present manuscript does contribute a meaningful and novel advance”, “the experiments are nontrivial and challenging”, and that “in general the broad claims of the paper appear sound and well justified”. We also thank the reviewer for all the helpful suggestions for us to improve the manuscript. Below we address the comments of the reviewer point by point.

Comment:

(1) The main claim of the paper is that the dual-type encoding scheme permits long-lived memory even during the ion-photon entanglement generation protocol. However, the two Yb ions are spaced by 12 μm , a significant distance, and it is not at all clear from the discussion how severe the crosstalk would be if the same $S_{\{1/2\}}$ encoding were used for the memory qubit. Can the authors include a measurement, or at least an estimate, of how severe the crosstalk-induced decoherence would be for the memory ion if a naive encoding in the same state were used, especially in light of the fact that the ions can in principle be more widely separated, e.g. by 100 μm in typical architectures? I.e., why was 12 μm chosen as the ion spacing? Given the $\sim 85\%$ fidelities for the memory reported, it is not clear that a memory ion in the $S_{\{1/2\}}$ state would undergo decoherence on this level even in the presence of the entanglement generation. Furthermore, since other trapped-ion systems do allow multiple photon scattering events on single ions even with other memory ions encoded (e.g. in Quantinuum’s systems, simply using some distance between measured and memory ions, e.g. PRA 104, 062440, 2021), this is a crucial comparison to make to understand how much the dual-type encoding really helps in this work. This is doubly important because the encoding used in this work entails significant infidelity in memory encoding/readout ($\sim 84\%$ mapping/storage fidelity). As such this seems a critical comparison to assess the value added by the demonstrated mapping.

Reply:

We thank the reviewer for this important comment. We divide the reviewer’s question into three parts: (1) How severe the crosstalk-induced decoherence would be if the memory ion were encoded as an S-qubit in this experiment? (2) Why was the ion spacing chosen as 12 μm in this experiment? (3) How is this dual-type encoding scheme compared with the QCCD and micromotion-hiding scheme used by

Quantinuum? Below we answer these three questions one by one.

(1) There are two types of crosstalk on the memory qubit during the ion-photon entanglement generation. The first is from the global 370 nm laser and the microwave to reset the state of the communication qubit before each entanglement attempt. In our previous work (Ref. [16]), we have already shown that this crosstalk is below 10^{-3} for the dual-type qubit encoding. On the other hand, it is clear that had we encoded the memory ion as an S-qubit, its state would have been completely destroyed by each reset operation. The second is the emitted photons from the communication ion hitting on the memory ion. It is widely known as a decoherence source in the ion trap quantum computing community (e.g. Ref. [5]) and is one of the motivations for the dual-species scheme in previous works (e.g. Ref. [12-14]). This decoherence was also observed in our previous work when some edge ions in a long chain were used for quantum memory while a few central ions were providing sympathetic laser cooling [Phys. Rev. A 106, 062617 (2022)]. To our knowledge, an analytical expression for this decoherence error has not been presented in the literatures. Here we obtain such an expression and add its derivation into Sec. II of Supplementary Information. From this formula, we estimate a crosstalk error $\epsilon = 6\%$ for an S-qubit at a distance of $d = 12 \mu\text{m}$ during the 100 ms entanglement generation. Such a decay in storage fidelity would be detected under our experimental precision, had we encoded the memory qubit in the S-type. On the other hand, by encoding the memory qubit into F-type, we are able to suppress this crosstalk and we observe no decay in storage fidelity up to statistical error bars $[(83.9 \pm 0.4)\% \rightarrow (83.6 \pm 0.8)\%]$ while generating ion-photon entanglement. Therefore, we conclude that our experiment demonstrates the capability of the dual-type qubit scheme to suppress the crosstalk error. We have added this into the main text and the Supplementary Information.

(2) Based on the above analysis, we can see that to demonstrate the effect of our dual-type qubit scheme, it is beneficial to use a smaller ion spacing, such that the crosstalk for the naïve encoding will be larger and that the advantage of the dual-type qubit encoding will be more significant. On the other hand, the smallest ion distance we can use is limited by the requirement of individual addressing of the two ions using 411nm laser. As we describe in the manuscript, in this experiment we use a focused

411nm laser with a beam waist radius of $4\mu\text{m}$, so that for accurate addressing of the memory qubit we choose an ion spacing of $12\mu\text{m}$. We would like to mention that this radius of $4\mu\text{m}$ is not a fundamental limit and we already obtain a smaller radius of $1.5\mu\text{m}$ in an on-going project, such that an ion spacing of $4\text{-}5\mu\text{m}$ can be achieved which is comparable to the other individually addressed trapped ion systems [e.g. Phys. Rev. Lett. 125, 150505 (2020); Nature 598, 281 (2021)].

(3) We would like to mention that in this work we do not claim the dual-type qubit scheme to be the only possible way to suppress the crosstalk error. Instead, we regard it as an appealing alternative way to the popular dual-species scheme because it can simplify the requirement to manipulate different ion species and can help solve the mass mismatch problem in sympathetic cooling. As for the QCCD scheme with micromotion-hiding [PRA 104, 062440 (2021)] demonstrated by Quantinuum as mentioned by the reviewer, we agree that it is another possible way to suppress the crosstalk error. However, for one thing, limited by the size of the electrodes, the distance between ions will need to be increased to tens to hundreds of micrometers to be able to displace them individually along the micromotion direction. Directly converting qubit types on site using the dual-type qubit scheme can be more convenient than transporting ions for this distance. For another, the QCCD scheme by itself achieves the advantage of simpler laser control and measurement at the cost of the more complicated transport operations. At this stage, we think that both the QCCD scheme and the scheme to directly perform quantum computing on a large ion crystal [see our previous theoretical proposals, Scientific reports 5, 8555 (2015) and Phys. Rev. A 103, 022419 (2021)] can be plausible candidates for a quantum computer node to be connected further by an ion-photon quantum network. Because a comparison between these different schemes is not the main purpose of this work, we choose not to add them into the manuscript.

Comment:

(2) Almost no detail is given to support the error budget described in the paragraph starting at line 184 — the authors should detail where the estimates for state detection/pumping, polarization, and misalignment come from, in the methods or supplementary information. As stated there is no basis for a reader to assess whether

these estimates are meaningful/accurate.

Reply:

We thank the reviewer for this helpful suggestion. We have added Sec. I in Supplementary Information to explain the ways we calibrate the experimental system and the ways we estimate these error sources. On the other hand, we would like to mention that given the overall infidelity above 10%, estimating its individual components can be inaccurate and that different parts may not add up together directly.

Comment:

(3) Similarly, in line 275, the 2.6% lower ion-photon entanglement achieved in the presence of the memory ion is explained as due to having switched from PMT to EMCCD — however, it is not clear why this should cause lower infidelity. Can the authors justify this due to higher effective dark count rates in the EMCCD, e.g., or is this due to crosstalk between ions as imaged? To claim that the error is due to the mechanism, the reason for the error should be detailed and at least estimated, or else the authors should revise to state that this is only suspected to be the cause for the increased error.

Reply:

We thank the reviewer for this important question. Our EMCCD gives larger detection error because it has lower detection efficiency than the PMT. There are mainly two reasons for this lower efficiency: a) Our EMCCD, without additional quantum enhancement, only has a quantum efficiency of about 23% at 370nm, which is lower than that of the PMT of about 30%. b) The PMT counts all the incoming photons, but when using the EMCCD we only count the pixels near the target ions to suppress the detection crosstalk. Therefore, to maintain the separation between the dark and the bright states, we increase the detection time from 300 μ s to 1 ms when switching from PMT to EMCCD. This longer detection time further leads to larger off-resonant pumping which causes the bright state to go outside the cyclic transition into the dark state, or reversely causing the dark state to become a bright state.

Following this suggestion and a relevant comment from Reviewer #3, we have added

a paragraph in Supplementary Information below Eq. (S1) to describe this mechanism and to provide the measured detection error for the bright and dark states. The theoretically estimated 3% increase in the infidelity bound is close to the measured 2.6%. Therefore we believe that this mechanism correctly explain the error source. Besides, since these errors are calibrated for a single ion using the selected number of pixels, we believe that the crosstalk does not play significant roles in this estimation.

Comment:

(4) In line 294, it is stated that the use of single ion entails simpler control techniques. I think this is appropriate with respect to sympathetic cooling and transport — however, with respect to the actual memory encoding and readout, the control is significantly more complex as evidenced in this work. This tradeoff should be acknowledged.

Reply:

We thank the reviewer for this helpful suggestion. We would like to mention that, the simplification in the dual-type qubit scheme is more on the hardware level because we do not need the hardware to manipulate different ion species and to adjust their fractions or positions. On the other hand, the arising complexity for encoding and readout is more on the software level by suitable sequences of the existing laser to convert the qubit types. (The 411 nm laser can be used for quantum gates and the 3432 nm laser can be used for repumping of ions from F levels, so they do not provide additional hardware cost for Yb-171 ions.) Therefore, we add the following sentence into the manuscript to acknowledge this tradeoff: “Admittedly, this simplification in hardware is achieved at the cost of more complicated software to convert the qubit type back and forth for the storage and readout, whose fidelity still needs to be improved.”

Comment:

(5) In general, measurement uncertainties, while likely well justified and correctly measured, are not clearly reported. Figs. 2-4 are missing statistical uncertainties on plotted points. No error bar is reported for the 87.9% ion-photon fidelity in line 174. In line 235, it is not entirely clear what is meant by “one standard deviation for the

total 9000 trials” — do the authors mean instead the standard error of the average fidelity inferred from these 9000 trials? In general, a more careful treatment of the measurement uncertainties would strengthen the presentation significantly.

Reply:

We thank the reviewer for this helpful suggestion. We have added statistical uncertainties and error bars to Figs. 2-4, and we compute the error bar for the entanglement fidelity bound as $(87.9 \pm 1.1)\%$ from those for the measured ion-photon correlations on page 2. As for line 235, yes, we mean the standard error of the average value. We have modified this sentence into “one standard error for the average of the total 9000 trials” to avoid confusion.

Comment:

A few more detailed minor suggestions for text:

- (1) line 21 — replace “ion trap has” with “trapped ions have”
- (2) line 42 — replace “scalization” with “scaling”
- (3) line 152 — since the fidelity was bounded but not measured by tomography, replace “measure” with “bound”. Similarly in the title to Fig. 2.
- (4) Line 266 — replace “300 data” with “300 trials”

Reply:

We thank the reviewer for these suggestions and have revised the manuscript accordingly.

Reviewer #3:**Comment:**

The work reported is a proof of principle demonstration of the use of the dual-type-state-manifold paradigm for generating ion-photon entanglement; this is done while simultaneously storing quantum information in a separate ion of the same species. Performing this operation with low crosstalk on the memory ion, also located in the same ion crystal, is one of the purported advantages of this paradigm, especially in light of the straightforward vibrational mode structure of a same-species ion crystal. Though there are many precursor experiments utilizing metastable states in trapped ions for protected memory, and in using spontaneous emission from ions to create ion-photon entanglement, the particular demonstration is novel in that it combines these two operations in the dual-type paradigm. Its publication should be useful for those in the field; improvements in the experiments must be undertaken before this can be a useful technique in trapped-ion quantum information processing and before a true limit to the dual-type technique can be reported, however, due to the low fidelities obtained here. The major drawback of the current work is that these relatively high infidelities limit the level at which crosstalk can be evaluated. Nonetheless, if this is clearly stated in the manuscript, I believe the work could be published.

Reply:

We thank the reviewer for the positive evaluation that “the particular demonstration is novel in that it combines these two operations in the dual-type paradigm” and that “its publication should be useful for those in the field”. We also agree with the reviewer that the current fidelity still needs to be improved in future works. However, as we describe in detail in the reply below and in our reply to Reviewer #2, the current fidelity already allows us to see the suppression of the crosstalk error in the dual-type qubit scheme compared with encoding all qubits in S-states. We believe that this is sufficient for a proof-of-principle demonstration, and as the reviewer says, after clearly stating this limitation in the manuscript, the work could be published.

Comment:

Summarizing improvements based on this general point and other detailed suggestions:

- 1) The authors should, in the discussion, provide an estimate for expected crosstalk

during the full experiment (memory during entanglement generation), and then highlight the improvements needed in the basic operations in order to be able to measure either a) crosstalk at this expected level, or if that level is much smaller than typical infidelities suggested for fault-tolerant operation (e.g. 10^{-4}), b) improvements required to get the crosstalk bound below such fault-tolerant infidelities.

Reply:

We thank the reviewer for this important suggestion. Following this suggestion and a relevant comment from Reviewer #2, we add Sec. II in Supplementary Information to theoretically estimate the crosstalk between two S-type qubits (without dual-type encoding) and between an S-qubit and an F-qubit (main results of this work). Theoretically, the crosstalk under dual-type encoding would be suppressed by a factor of about 10^{-12} , which is far below the fault-tolerant threshold and can be safely neglected. On the other hand, although our current experimental precision is not able to resolve such a small crosstalk (and not likely in the near future), our error bar below 1% in measuring storage fidelity is already sufficient to show the suppression of the crosstalk compared with the 6% crosstalk error from the scattered photons and even larger crosstalk from the global beams had we encoded into two S-type qubits.

As for the reviewer's question about what improvements are needed to verify a crosstalk below the fault-tolerant threshold, say, 10^{-4} , we think the key factor is not the fidelity of all the operations, but the precision of the fidelity measurement. The crosstalk error is computed as the difference between the storage fidelity with/without ion-photon entanglement generation. Therefore, to be able to measure a fidelity change by 10^{-4} caused by the crosstalk, we will need about $M = 10^8$ experimental trials since the statistical error scales as $1/\sqrt{M}$. Because each experimental trial involves quantum storage of 200 ms, the overall experimental time would be about one year and would impose high requirement on the stability of the system. A more practical scheme is to increase the number of qubits. Suppose we have 100 ions generating ion-photon entanglement simultaneously, and we measure the total crosstalk on a nearby memory qubit, then a precision of 1% will be sufficient to bound the crosstalk error per ion to the 10^{-4} level. We have added this into Discussion.

Comment:

2) Wording suggestion for the title: "Realization of a crosstalk-avoided quantum network node using dual-type qubits of the same ion species."

Reply:

We thank the reviewer for this suggestion and have modified the title accordingly.

Comment:

3) Lines 59-61: the motivation for dual-type is a little confusing here when discussing the challenges of dual-species operation--perhaps the authors mean to say that the proportion of ions of a particular species and the ion order are hard to control? On the contrary, it seems that the recent work of Quantinuum has shown that several multi-ion, dual-species crystals of a particular ion order can be accomplished without much difficulty. I would remove this statement and focus on the challenges of cooling all the vibrational modes in a dual-species crystal, as is mentioned in the next sentence.

Reply:

We thank the reviewer for this suggestion. We have corrected the word "proportion" to avoid confusion. First we would like to mention that, by this sentence, we mean not only the QCCD scheme use by e.g. Quantinuum, but also the scheme to directly perform quantum computing on a whole large ion crystal with two different ion species locating on different sites. Both these schemes require accurate control of the proportion and the location of individual ion species, and for the QCCD scheme, even the location of individual ion pairs. For a large ion crystal, it is clear that the dual-type qubit scheme is much simpler than the dual-species scheme because all the ions are identical particles and we can arbitrarily assign the two qubit types simply by laser pulses.

We agree with the reviewer that the recent work of Quantinuum [arXiv:2305.03828] shows important progress for the QCCD scheme, but we think it is not "accomplished without much difficulty". Instead, it is the result of more than 20 years' research since the initial idea of the QCCD architecture (Ref. [5]), and it takes nearly 100 researchers (number of authors of arXiv:2305.03828) to work for years. Besides, the current

Quantinuum H2 device still uses a 1D (ring) topology. There can still be technical challenges to accurately control the transport across junctions when extending to larger 2D structures, although the basic function has already been demonstrated [Advanced Quantum Technologies 3, 2000028 (2020)].

Therefore, we believe that despite the important progress of Quantinuum, our statement about the difficulty to manipulate multiple ion species can still be a valid motivation for the dual-type qubit scheme.

Comment:

4) The authors should describe the limitations to crosstalk avoidance presented by the use of focused laser beams for individual addressing, one of the drawbacks of dual-type protocols for qubits in the same ion crystal. The beam waists for the addressing beams quotes here must be comparable to the inter-ion spacing, leading to accidental crosstalk in the mapping operations. This deserves some discussion, as it could present a practical limitation, particularly in terms of attainable trap-potential strength, or to speed if the potential must be weakened to perform this operation.

Reply:

We thank the reviewer for this suggestion. Because the 411 nm laser is within the typical wavelength range (from ultraviolet to red) for individual addressing of quantum gates on ionic qubits, we believe that its beam waist will not be a fundamental limit. In fact, in an on-going project, our group already obtains a smaller radius of 1.5 μm for the 411 nm laser, such that an ion spacing of 4-5 μm can be achieved without significant crosstalk error, which is comparable to the other individually addressed trapped ion systems [e.g. Phys. Rev. Lett. 125, 150505 (2020); Nature 598, 281 (2021)].

Comment:

5) How is state initialization of one ion in a two-ion crystal done here? Are there challenges due to scattered photons affecting the other ion in this case?

Reply:

The initialization of the memory qubit is realized by global Doppler cooling, optical

pumping and microwave pulse. After optical pumping, the steady state is that both ions are in the dark state, so that there is no crosstalk due to scattered photons. Such global operations clearly introduce unnecessary operations on the communication qubit as well, but it is not a source of error because later the communication qubit will be initialized again in each attempt to generate ion-photon entanglement. After the initialization of the memory qubit in the S-type, we further use individual 411 nm and global 3432 nm laser pulses to convert it to the F-type. Then we initialize the communication ion again using global Doppler cooling, optical pumping and microwave pulse for the following ion-photon entanglement generation, but this time, since the memory qubit is already on the $F_{7/2}$ levels, it will not be affected by these global beams.

Comment:

6) Why is the qubit brought from the F state to the S state for MW operations? Can these not be done in the F state to avoid the infidelity of two state-mapping operations?

Reply:

The reviewer is correct that to solely demonstrate the crosstalk-avoided quantum memory during the generation of ion-photon entanglement, we could initialize and read out the memory qubit in the F state with all the required tools demonstrated in our previous work (Ref. [16]). If this is the case, then only the relative phase change between the F-qubit and its driving microwave will matter, and there will be no need to bring the qubit back to the S state for spin echo. However, to demonstrate the full advantage of the dual-type qubit scheme, we further want to show the capability to convert the qubit types and to maintain the coherence between the two types after long-time storage. Therefore we choose to initialize and read out the memory qubit in the S state and only move it to the F state for protection against crosstalk error during the ion-photon entanglement generation. In this case, the relative phase change between the S-qubit and its driving microwave, as well as the small frequency shift in the two components of the bichromatic 411 and 3432 nm laser, will all contribute to the dephasing of the memory qubit. Therefore we bring the qubit back to the S state for spin echo, and we design the pulse sequence in such a way that a constant

frequency shift in any of these components can be cancelled.

Comment:

7) Fig 1 suggestion: use the same colors for the laser beams corresponding to each transition in parts a, b, and e to allow easier comprehension by the reader. Also, label the laser beams with wavelengths in part e.

Reply:

We thank the reviewer for this suggestion and have revised Fig. 1 accordingly.

Comment:

8) It is not stated why the EMCCD-based detection adds more errors--this should be explained, and any fundamental or technical limitations in the number of measurable ions due to this effect should be described.

Reply:

We thank the reviewer for this helpful suggestion. Our EMCCD gives larger detection error because it has lower detection efficiency than the PMT. There are mainly two reasons for this lower efficiency: a) Our EMCCD, without additional quantum enhancement, only has a quantum efficiency of about 23% at 370nm, which is lower than that of the PMT of about 30%. b) The PMT counts all the incoming photons, but when using the EMCCD we only count the pixels near the target ions to suppress the detection crosstalk. Therefore, to maintain the separation between the dark and the bright states, we increase the detection time from 300 μ s to 1 ms when switching from PMT to EMCCD. This longer detection time further leads to larger off-resonant pumping which causes the bright state to go outside the cyclic transition into the dark state, or reversely causing the dark state to become a bright state.

Following this suggestion and a relevant comment from Reviewer #2, we have added a paragraph in Supplementary Information below Eq. (S1) to explain this reason.

Comment:

9) One of the fidelity limitations was due to ion heating since sympathetic cooling was not used. It's not clear that this would be a major limitation unless the ions were

heated to a particularly high temperature; what was the heating rate in this experiment, and was it the main limitation to mapping operation infidelity? Or was there a limit fur to heating from the photon entanglement attempts causing recoil? If so, how could this be fixed? (Is a third ion required?) This is not well explained in the paper.

Reply:

We thank the reviewer for this important question. First we would like clarify that we do perform sympathetic cooling in this experiment. What we don't have yet is the sympathetic sideband cooling (or other sub-Doppler cooling) to cool the phonon number below one. Therefore, heating will not be a big problem in this experiment, and the ions will roughly stay at the Doppler temperature (we measure an average phonon number of about 16, which is 4 times the Doppler cooling limit).

Second, this phonon number, although not particularly high, can be an important infidelity source in the qubit-type conversion. For a given phonon number of n , it is well-known that the carrier Rabi frequency will be modified by a factor of $e^{-\eta^2/2}L_n(\eta^2)$ where η is the Lamb-Dicke parameter, and $L_n(x)$ is the Laguerre polynomial (see, e.g. Ref. [24]). In this experiment we have $\eta \approx 0.054$ for the 411 nm laser and a trap frequency of $2\pi \times 2.4$ MHz. For an average phonon number of \bar{n} under thermal distribution, the standard deviation is $\sqrt{\bar{n}(\bar{n} + 1)}$, therefore we estimate the fluctuation in the carrier Rabi frequency as $\sqrt{\bar{n}(\bar{n} + 1)}\eta^2$ where we use $L_n(\eta^2) \approx 1 - n\eta^2$ when $n\eta^2 \ll 1$. For the qubit-type conversion, we want a π pulse for the 411nm laser, and the error will be $\cos^2 \pi \left(1 \pm \sqrt{\bar{n}(\bar{n} + 1)}\eta^2\right)/2 \approx (\pi/2)^2 \bar{n}(\bar{n} + 1)\eta^4$. For a round-trip conversion, we have two 411 nm π pulses, so that the error is $2(\pi/2)^2 \bar{n}(\bar{n} + 1)\eta^4 \approx 1\%$. This explains about half of the increase in the round-trip conversion fidelity from 1% in our previous work with sideband cooling (Ref. [16]) to 3% here with only Doppler cooling. We suspect that an additional factor of two can appear when we perform two nearby conversions in the spin echo sequence separated barely by a microwave π pulse, such that the pulse area error adds up coherently. If this is the case, we can insert a random phase shift between two adjacent 411 nm conversion pulses in the future to suppress this coherent error.

Finally, there can be several ways to solve this problem and to improve the conversion fidelity. As we discuss in the manuscript, one possible way is to add a third ion to continuously provide sympathetic sideband cooling. Another possibility is to use just two ions, but pause the entanglement generation sequence once every tens of attempts to provide sideband cooling. Given our heating rate of about 100 phonons/s and the entanglement generation rate of about $1.6 \times 10^4 \text{ s}^{-1}$, the phonon number can be controlled below one to allow high-fidelity qubit-type conversion.

We have added the above analysis and discussions into the manuscript and Sec. III of Supplementary Information.

Comment:

10) The authors discuss a future approach not requiring the combination of two Zeeman levels coherently--was this a limit to fidelity in the current scheme? If so, the authors should describe that. Also, in this same part of the paper, the authors also state that state detection could be improved using electron shelving--how could this be implemented in the current scheme, since the F state is being used and the D state is not long lived? The authors should expand this point with another sentence or two describing how this could be done without involving undue complications.

Reply:

We thank the reviewer for this helpful suggestion. We think the combination of the two Zeeman levels is one of the important error sources in this experiment, and contribute to about 6% infidelity in the ion-photon entanglement (about 1/2 of the total infidelity). We have added an analysis for the possible sources of errors in Sec. I of Supplementary Information. In comparison, when using a scheme without the need to combine two Zeeman levels, the jitter time and the dephasing will only affect the off-diagonal basis, while the diagonal basis (population) will be more robust.

As for the reviewer's question about how electron shelving can be realized given the F levels are being used, we would like to mention that one advantage of the dual-type qubit scheme is that the qubit type can be converted on demand individually. (Admittedly, the current conversion fidelity still needs to be improved, but as we describe in the above replies, we believe there is no fundamental limit.) To achieve

electron shelving for the measurement of S-qubits without affecting other stored F-qubits, one possible way is to use individual 411 nm laser and a weak global 3432 nm laser to shelve to other Zeeman levels of $F_{7/2}$ not used for encoding qubits. Another possibility is to use a sequence of “global 3432 π pulse”-“individual 411 π pulse”-“global 3432 π pulse” to selectively convert the qubit type for the targeted ions without affecting the other ions (apart from the imperfect π pulses). We have added these into Discussion.

Comment:

11) The English should be proofread and edited throughout. Also, the hyphenation breaks across lines are very atypical, e.g. line 287, etc.

Reply:

We thank the reviewer for this suggestion and have proofread and revised the manuscript. As for the hyphenation breaks across lines, they are automatically generated by the latex template, so we guess this will not be a problem in the editing process.

Comment:

If the authors suitably address my concern listed here, and if the other reviewer(s) agree, I believe the work could be published.

Reply:

We thank the reviewer for this positive attitude for publication. We hope that we have addressed all the concerns of the reviewer above.

REVIEWER COMMENTS

Reviewer #2 (Remarks to the Author):

The authors have responded to earlier comments from me and the other reviewers thoroughly and in a way that I think significantly strengthens the paper. I don't have further comments, and I recommend publication of the revised manuscript.

Reviewer #3 (Remarks to the Author):

Comments on resubmission by Feng et al.

The authors have adequately responded to some of my concerns and those of the other referees, but not all. I feel there is still some revision that is required. Detailed comments/questions follow:

1) In the authors rebuttal to referee 2, I think they are missing the point that referee is making about the naïve encoding (both ions in the S1/2) for ions 12 μm apart. The authors estimate a 6% error in a naively encoded qubit in a neighboring ion 12 μm away during entanglement generation as performed here. But the infidelity in memory is $\sim 16\%$. I think the point to take away here, and what I believe referee 2 was getting at, is that the dual-type encoding as shown here is significantly worse than the naïve encoding would be in the same experimental setup of same-species ions 12 μm apart. The authors must therefore explain in the paper why this demonstration is sufficient to promote the use of dual-type encodings. The authors need to make that case, since no improvement over naïve standard encoding of same-species ions is shown here.

2) The authors also don't adequately address referee 2's comment about simpler control techniques in dual-type versus dual-species encodings. The authors say that, when going to the dual-type encoding, the simplicity primarily comes in hardware, and the added complexity primarily comes in the software (e.g. mapping from ground to metastable levels). However, this neglects the requirement for individual-ion addressing in dual-type encoding for these transfer operations that use the same transitions in all ions. Such individual addressing does point to specific hardware tradeoffs like ion spacing (as

mentioned above) and/or tight focusing of beams. This tradeoff is important and should be mentioned in addition to the mention of “software” complexity.

3) [Now on to referee 3 comments/rebuttals] On my point (1) in my previous report, about the precision at which the crosstalk error can be determined, the authors respond that 10^8 experiments would be required to get a smaller uncertainty, but this would take prohibitively long with their current setup and experimental sequence. Alternatively, they state that they'd need 100 communication ions and one memory ion to get a larger signal for the crosstalk. This seems experimentally challenging as well, since the authors do not explain in what type of trap they would be able to achieve a configuration with 100 ions 12 μm away from a single memory ion. Or maybe they are suggesting 100 communication-memory ion pairs operated simultaneously? This seems challenging as well, at least with the team's current capabilities. Since neither of these measurements could occur soon, it seems that to truly determine the crosstalk error at a level that would convince the community that this encoding can work, e.g. 10^{-4} or so, we must wait quite a while. Since I feel that this is an important consideration that any reader trying to evaluate the viability of the dual-type encoding would have, the authors should state in the paper that they cannot put a quantum-computing-relevant bound on the achievable crosstalk in their scheme in the near term, even though it looks good at low precision.

4) On my point (3) in my previous report, about the motivations for single-species approaches, the authors missed the main point. Prior work with the QCCD architecture, like Quantinuum's, have demonstrated that, contrary to the authors' claims in the paper and rebuttal, it is not difficult to load multiple dual-species ion crystals as needed, in the right order and proportion. This is done regularly in systems with up to a few tens of ions. The fact that it has taken the field 20 years of research to get to this point is irrelevant—one could use that argument about anything that has been accomplished based on prior work, but is nonetheless demonstrated and widely applicable, e.g. laser cooling, electron shelving-based detection, even single-ion trapping. I would strongly suggest removing the proportion/order motivation. The other statements about challenges with cooling in dual-species crystals are the points to focus on. (The authors also bring up potential challenges with dual species crystals in junctions, but Quantinuum has also recently shown fast

shuttling of dual-species crystals through junctions with very low excitation, so this also does not seem to be a problem.)

Reply to Reviewers

Reviewer #2:

Comment:

The authors have responded to earlier comments from me and the other reviewers thoroughly and in a way that I think significantly strengthens the paper. I don't have further comments, and I recommend publication of the revised manuscript.

Reply:

We are glad to know that the reviewer is satisfied with our previous reply, and we thank the reviewer for recommending publication of our revised manuscript.

Reviewer #3:

Comment:

Comments on resubmission by Feng et al.

The authors have adequately responded to some of my concerns and those of the other referees, but not all. I feel there is still some revision that is required. Detailed comments/questions follow:

Reply:

We thank the reviewer for carefully reading through our reply to all the reviewers, and for the helpful suggestions to improve the manuscript. Below we address the comments of the reviewer point by point.

Comment:

1) In the authors rebuttal to referee 2, I think they are missing the point that referee is making about the naïve encoding (both ions in the $S1/2$) for ions 12 μm apart. The authors estimate a 6% error in a naively encoded qubit in a neighboring ion 12 μm away during entanglement generation as performed here. But the infidelity in memory is $\sim 16\%$. I think the point to take away here, and what I believe referee 2 was getting

at, is that the dual-type encoding as shown here is significantly worse than the naïve encoding would be in the same experimental setup of same-species ions 12 μm apart. The authors must therefore explain in the paper why this demonstration is a sufficient to promote the use of dual-type encodings. The authors need to make that case, since no improvement over naïve standard encoding of same-species ions is shown here.

Reply:

We would like to clarify that, in our previous reply to Reviewer #2, we consider two types of crosstalk errors on the memory qubit: the emitted photons from the communication qubit which is intrinsic in any ion-photon entanglement generation process, and all the other auxiliary operations like Doppler cooling, optical pumping and microwave pulses on the communication qubit. The value of 6% quoted by the reviewer only covers the intrinsic photon emission in the ion-photon entanglement generation. However, when both types of crosstalk errors are considered, our demonstration of the dual-type qubit scheme does have significantly better performance compared with the naïve encoding. Actually, a single global microwave pulse on the communication qubit would have $O(1)$ crosstalk error on a naïvely encoded memory qubit, while we have $1.6 \times 10^4 \text{s}^{-1} \times 100 \text{ms} = 1600$ such pulses in the experimental sequence in Fig. 4. Therefore, we believe that the reviewer's statement that "the dual-type encoding as shown here is significantly worse than the naïve encoding" is not the case for our experiment.

Also note that, although we call these auxiliary operations as "not intrinsic", this does not mean they can be easily avoided. For example, in this experiment we use global Doppler cooling and optical pumping beams for the communication qubit, which would have caused large crosstalk errors had we encoded the memory qubit in the S-type. In principle, one could use a focused laser beam for these operations, but still it is inevitable (and necessary) to generate a few scattered photons in this process in each reset-generation cycle, which would increase our computed crosstalk error of 6% by several times. (Note that the estimation of 6% comes from one photon scattering per reset-generation cycle). Furthermore, the global microwave pulses are difficult to be applied selectively on the communication qubit. We have emphasized these in the revised Supplementary Information.

Comment:

2) The authors also don't adequately address referee 2's comment about simpler control techniques in dual-type versus dual-species encodings. The authors say that, when going to the dual-type encoding, the simplicity primarily comes in hardware, and the added complexity primarily comes in the software (e.g. mapping from ground to metastable levels). However, this neglects the requirement for individual-ion addressing in dual-type encoding for these transfer operations that use the same transitions in all ions. Such individual addressing does point to specific hardware tradeoffs like ion spacing (as mentioned above) and/or tight focusing of beams. This tradeoff is important and should be mentioned in addition to the mention of "software" complexity.

Reply:

We thank the reviewer for this suggestion. We have added: "Also, our scheme needs individual addressing with focused laser beams to convert the desired qubits selectively. This is already available in many current multi-ion quantum computers for individually addressed quantum gates."

Comment

3) [Now on to referee 3 comments/rebuttals] On my point (1) in my previous report, about the precision at which the crosstalk error can be determined, the authors respond that 10^8 experiments would be required to get a smaller uncertainty, but this would take prohibitively long with their current setup and experimental sequence. Alternatively, they state that they'd need 100 communication ions and one memory ion to get a larger signal for the crosstalk. This seems experimentally challenging as well, since the authors do not explain in what type of trap they would be able to achieve a configuration with 100 ions 12 μm away from a single memory ion. Or maybe they are suggesting 100 communication-memory ion pairs operated simultaneously? This seems challenging as well, at least with the team's current capabilities. Since neither of these measurements could occur soon, it seems that to truly determine the crosstalk error at a level that would convince the community that this encoding can work, e.g. 10^{-4} or so, we must wait quite a while. Since I feel that this is an important consideration that any reader trying to evaluate the viability of the

dual-type encoding would have, the authors should state in the paper that they cannot put a quantum-computing-relevant bound on the achievable crosstalk in their scheme in the near term, even though it looks good at low precision.

Reply:

We thank the reviewer for this comment. We would like to divide our answer into two parts: (1) What scheme we were considering in the previous reply to bound the crosstalk error to 10^{-4} level, and (2) is this bound of 10^{-4} necessary for the future fault-tolerant quantum computing, given that it is an accumulated memory error over long storage time.

First, by “having 100 ions generating ion-photon entanglement simultaneously” in our previous reply, we mean the first understanding of the reviewer, namely ~ 100 communication ions and one memory ion. The reviewer is correct that it is impossible to hold all these communication ions at the same distance of $d=12\mu\text{m}$ from the memory ion, but if one accepts the scaling analysis that the crosstalk error roughly scales as $1/d^2$, then one can still estimate the overall crosstalk error from the ions at different distances. What we had in mind when we mentioned the 100 ions was a 2D ion crystal. Currently in our lab we can already obtain a 2D crystal with hundreds of ions, and recently Innsbruck also reported a 2D crystal with above 100 ions [PRX Quantum 4, 020317 (2023)], and tens of ions at University of Washington [Phys. Rev. A 105, 023101 (2022)]. With our typical ion spacing of about $4.5\mu\text{m}$ in the 2D crystal, and assuming a uniform triangular lattice for simplicity (6 nearest neighbors at the distance of d , and about $6n$ ions at the distance of roughly nd in the n -th shell), we can estimate that a crystal of tens of ions will be sufficient to give about 100 times the crosstalk error from a single communication qubit at a distance of $12\mu\text{m}$. We agree with the reviewer that our wording of “100 ions generating ion-photon entanglement simultaneously” can be confusing. We have revised it into “A more practical scheme would be to have a large ion crystal with smaller ion distances to generate ion-photon entanglement simultaneously, and to measure the total crosstalk error on a memory qubit at the center of the crystal.”

Second, we would like to argue that, for fault-tolerant quantum computing, it is not required for this crosstalk error to be far below the commonly used definition of

fault-tolerant error correction threshold (although our theoretical analysis suggests that it is indeed far below). Note that the commonly used definition of fault-tolerant threshold applies to individual quantum gates and measurements which are used in the error correction process, and the reason why their error rates must be low is to prevent generating more errors from correcting the existing ones. On the other hand, the crosstalk error we measure in this experiment is the error rate on the memory qubit during 100ms entanglement generation on the communication qubit. This is about one half of the crosstalk error per successful ion-photon entanglement whose time scale is 200ms. Note that during this 100ms time period, there are in total 1600 attempts to generate ion-photon entanglement, and for the future fault-tolerant quantum computing, we can insert quantum error correction on the memory qubit between any of these attempts. For example, if we perform error correction once every 100 entanglement attempts, then the error rate on the memory qubit will be 16 times smaller than the crosstalk error we measure. Furthermore, this type of crosstalk error is more like the “memory error” rather than the “gate error” or the “measurement error” as mentioned above, and is less severe in fault-tolerant quantum error correction. For example, it is known that for the surface code, under perfect error syndrome measurement, a memory error rate of 11% can be tolerated for both the bit-flip and the phase-flip errors [J. Math. Phys. 43, 4452–4505 (2002)]. Therefore, we believe that bounding the crosstalk error to be below 10^{-4} will not be necessary even for the future fault-tolerant quantum computing. We have added “Also note that this crosstalk error is more like a memory error rather than the gate error or the measurement error in quantum error correction, and is thus less severe for fault-tolerance. For example, for the surface code it is known that a memory error of 11% for both the bit-flip and the phase-flip errors can be corrected” to clarify this point.

Comment:

4) On my point (3) in my previous report, about the motivations for single-species approaches, the authors missed the main point. Prior work with the QCCD architecture, like Quantinuum’s, have demonstrated that, contrary to the authors’ claims in the paper and rebuttal, it is not difficult to load multiple dual-species ion crystals as needed, in the right order and proportion. This is done regularly in systems

with up to a few tens of ions. The fact that it has taken the field 20 years of research to get to this point is irrelevant—one could use that argument about anything that has been accomplished based on prior work, but is nonetheless demonstrated and widely applicable, e.g. laser cooling, electron shelving-based detection, even single-ion trapping. I would strongly suggest removing the proportion/order motivation. The other statements about challenges with cooling in dual-species crystals are the points to focus on. (The authors also bring up potential challenges with dual species crystals in junctions, but Quantinuum has also recently shown fast shuttling of dual-species crystals through junctions with very low excitation, so this also does not seem to be a problem.)

Reply:

We thank the reviewer for this suggestion and have removed this sentence.

REVIEWER COMMENTS

Reviewer #3 (Remarks to the Author):

I have only two remaining issues with the authors' current response and current manuscript version, though I think these issues are important as they get to the motivation and benefit of adopting the new encoding scheme espoused by the authors. If these can be addressed, the paper can proceed.

In their reply to my comment (1), the authors say that their experiment does indeed show an improvement over the naïve encoding of two same-species ions 12 μm apart because control fields as currently applied lead to large cross-talk errors. But unlike the 6% due to spontaneous scattering, the control crosstalk is not a fundamental source of error--one is not required to perform MW operations; focused-laser Raman operation could provide the same functionality. The authors also state that there are other sources of scattered photons from state reset and ion cooling--these ARE fundamental sources of error when comparing the two schemes. The authors should carefully calculate what the fundamental sources of error in the naïve scheme are and compare their achieved results to those to determine if the dual-type scheme can provide an improvement.

In their reply to my comment (3) on what level of fidelity (and precision on fidelity) is useful for this demo, the authors claim that even an 11% error in memory would be acceptable. This bound assumes perfect syndrome extraction, which is completely impractical, and goes against the whole notion of fault tolerance. Quantum error correction that is fault tolerant requires the ability to handle errors during syndrome determination as well, since these are made up of physical operations just like the others. This is why nobody in the community assumes that 11% is a reasonable infidelity to aim for in producing a quantum computer. And indeed, if 11% were fine, we'd already have the required fidelity for a large-scale quantum computer in several physical modalities. But the fact is that we don't, and those in the field have chosen much smaller infidelities to aim for, knowing that error rates must be far below the roughly 1% value given for the surface code in order to avoid impractical amounts of overhead. Though a real threshold is not precisely known, and depends on QEC architecture, the authors should base their target value on reasonable values derived from

known codes. 11% does not fit this categorization, and is not an acceptable target to those in the field. The authors need to pick a reasonable target infidelity and compare their results to that number in order to determine the prospects for high-fidelity operation under the dual-type protocol as described here.

Reply to Reviewers

Reviewer #3:

Comment:

I have only two remaining issues with the authors' current response and current manuscript version, though I think these issues are important as they get to the motivation and benefit of adopting the new encoding scheme espoused by the authors. If these can be addressed, the paper can proceed.

Reply:

We thank the reviewer for the helpful suggestions. Below we address the comments of the reviewer point by point.

Comment:

In their reply to my comment (1), the authors say that their experiment does indeed show an improvement over the naïve encoding of two same-species ions 12 um apart because control fields as currently applied lead to large cross-talk errors. But unlike the 6% due to spontaneous scattering, the control crosstalk is not a fundamental source of error--one is not required to perform MW operations; focused-laser Raman operation could provide the same functionality. The authors also state that there are other sources of scattered photons from state reset and ion cooling--these ARE fundamental sources of error when comparing the two schemes. The authors should carefully calculate what the fundamental sources of error in the naïve scheme are and compare their achieved results to those to determine if the dual-type scheme can provide an improvement.

Reply:

We thank the reviewer for this suggestion. As shown below, we estimate there to be at least $k = 11$ fundamental photon scattering due to optical pumping in each entanglement attempt. This will increase our estimation of the fundamental crosstalk error for naïve encoding from $\epsilon = 6\%$ for one photon per entanglement attempt to $1 - (1 - \epsilon)^{k+1} \approx 50\%$ for the $k + 1$ photons. We have added the following

calculation into Supplementary Information: “Furthermore, as shown in Fig. 1c of the main text, before each ion-photon entanglement attempt, we initialize the communication ion by Doppler cooling and optical pumping. During the $40\mu\text{s}$ Doppler cooling, hundreds of photons will be scattered, but how frequently we need to perform the Doppler cooling depends on the heating rate and thus has room for improvement. In comparison, the photon scattering during optical pumping is more intrinsic. Since the $^{171}\text{Yb}^+$ ion has $1/3$ probability to decay to the dark state $|\downarrow\rangle$ after each spontaneous emission, we can estimate a state preparation infidelity of about $(1 - 1/3)^k$ after the emission of k photons. In other words, about $k = 11$ photons will be necessary for a state preparation error of 1%. Therefore, we estimate an intrinsic crosstalk error of $1 - (1 - \epsilon)^{k+1} \approx 50\%$, had we encoded the memory qubit in the S state.” Note that this 50% fundamental crosstalk error for naïve encoding is not only higher than the 1% upper bound on our measured crosstalk error under dual-type encoding due to the experimental precision (the actual value can be orders of magnitude smaller), but also higher than the overall measured storage infidelity of about 16% with all the technical errors like SPAM errors and conversion errors included. Therefore our dual-type scheme does provide an improvement. We have emphasized this in the main text: “In comparison, without the dual-type qubit encoding, the memory qubit would have experienced much larger crosstalk error which would have been detected under our experimental precision, and would have even exceeded our overall measured storage infidelity of 16% with the technical errors included (see Supplementary Information for details).”

Comment:

In their reply to my comment (3) on what level of fidelity (and precision on fidelity) is useful for this demo, the authors claim that even an 11% error in memory would be acceptable. This bound assumes perfect syndrome extraction, which is completely impractical, and goes against the whole notion of fault tolerance. Quantum error correction that is fault tolerant requires the ability to handle errors during syndrome determination as well, since these are made up of physical operations just like the others. This is why nobody in the community assumes that 11% is a reasonable infidelity to aim for in producing a quantum computer. And indeed, if 11% were fine,

we'd already have the required fidelity for a large-scale quantum computer in several physical modalities. But the fact is that we don't, and those in the field have chosen much smaller infidelities to aim for, knowing that error rates must be far below the roughly 1% value given for the surface code in order to avoid impractical amounts of overhead. Though a real threshold is not precisely known, and depends on QEC architecture, the authors should base their target value on reasonable values derived from known codes. 11% does not fit this categorization, and is not an acceptable target to those in the field. The authors need to pick a reasonable target infidelity and compare their results to that number in order to determine the prospects for high-fidelity operation under the dual-type protocol as described here.

Reply:

We totally agree with the reviewer that the 11% threshold for the memory error assumes perfect syndrome extraction, and that it is NOT what the community (including our group) is aiming for. We have removed this number to avoid confusion in the revised manuscript. What we meant in our previous reply was that the memory error (accumulated over 100ms when stored on F states) should be treated differently compared with the gate errors and the SPAM errors (below a few milliseconds). Actually, Ref. [27] already considers these two types of errors: bit-flip and phase error rate p on individual qubits, which corresponds to our memory error, and measurement error rate q , which also includes gate errors during the syndrome measurements. When $q = 0$, one gets the above threshold $p_c \approx 11\%$ for the memory error under perfect syndrome measurement. The reviewer is correct that when q increases, the threshold p_c will decrease. In particular, when assuming $p = q$, one recovers the commonly quoted threshold around $p_c \approx 1\%$. On the other hand, one can expect that when q is sufficiently lower than the 1% threshold, the corresponding threshold for p will approach 11%. Similar phenomenon has also been observed in a related work using the surface code for quantum communication [Phys. Rev. Lett. 104, 180503 (2010)]: “Permitting repeaters to have a nonzero local gate error rate p_g will only have significant impact if it is close to the threshold error rate of approximately $p_g^{\text{th}} = 0.75\%$. An error rate 1 or 2 orders of magnitude below this will not significantly change the above results”. We have also simulated the performance of the surface code under different values of p and q using the Stim

package [arXiv:2103.02202]. As shown in the figure below, we fix the error rates of quantum gates, state initialization and measurement all to be $q = 0.1\%$, and consider two possible memory error rates $p = 5\%$ (blue) and $p = 1\%$ (orange). In both cases, the logical error rate decays exponentially as the code distance increases, which confirms that these memory error rates are already below the corresponding threshold under the fixed $q = 0.1\%$.

Nevertheless, we completely agree with the reviewer that further suppressing the error rate to be far below the threshold will surely be beneficial and will help save the overhead for quantum error correction. As described in our previous reply, we have added discussions in the Discussion section of the main text about how to experimentally confirm a low crosstalk error below 10^{-4} using a large ion crystal by measuring the total crosstalk error from multiple communication ions. Here, to avoid confusion of the readers whether we are satisfied with the 11% threshold for memory error, we have removed this number, and have replaced it by “Nevertheless, it is always desirable to achieve and to verify lower crosstalk errors to save the overhead for large-scale quantum error correction”.

REVIEWERS' COMMENTS

Reviewer #3 (Remarks to the Author):

The authors have sufficiently addressed my concerns, but I would request that the authors please point out in the manuscript that the memory error they achieve here must be reduced significantly in order to make this approach viable for fault-tolerant quantum computing, and they should suggest how this can be achieved. As the graph included in their reply shows, a linear increase in error rate for physical operations including measurement leads to an exponential penalty in resources (i.e. a significant increase in code distance) to achieve a given error rate. The reader will require this caveat.

Reply to Reviewers

Reviewer #3

Comment:

The authors have sufficiently addressed my concerns, but I would request that the authors please point out in the manuscript that the memory error they achieve here must be reduced significantly in order to make this approach viable for fault-tolerant quantum computing, and they should suggest how this can be achieved. As the graph included in their reply shows, a linear increase in error rate for physical operations including measurement leads to an exponential penalty in resources (i.e. a significant increase in code distance) to achieve a given error rate. The reader will require this caveat.

Reply:

We are glad to know that the reviewer is satisfied with our previous reply. In the previous version, we already had a paragraph in the Discussion section about how to improve the conversion fidelity, which is the main source of memory error in this experiment. Following the suggestion of the reviewer, we state in the beginning of this paragraph explicitly about the necessity to improve the overall memory fidelity: “The current storage infidelity of the memory qubit is about 16% over the 200ms storage time, which must be significantly improved for the future fault-tolerant quantum computing. As mentioned above, among the 16% infidelity, only about 3% comes from the measured coherence time, while the dominant sources are the conversion infidelity between the S-qubit and the F-qubit, and the SPAM error using EMCCD.” Then, after discussing the source of the conversion infidelity and the way to improve it with sideband cooling and sympathetic cooling, we further discuss how the SPAM error can be suppressed by electron shelving: “As for the SPAM error from the EMCCD detection, it can be improved by electron shelving”. Previously, later in the manuscript we had a discussion about how to combine the electron shelving technique with the dual-type qubit scheme. We have moved these sentences into this paragraph. Finally, for the crosstalk error from the communication qubit, we already had a discussion in the next paragraph about how to provide a better upper bound of 10^{-4} in future experiments in the previous version.